# Effects of intestinal parasite infection on hematological profiles of pregnant women attending antenatal care at Debre Markos Referral Hospital, Northwest Ethiopia: Institution based prospective cohort study

**Gebreselassie Demeke**[1]*, **Getachew Mengistu**[1], **Abtie Abebaw**[1], **Milkiyas Toru**[1], **Molla Yigzaw**[1], **Aster Shiferaw**[2], **Hylemariam Mihiretie Mengist**[1], **Tebelay Dilnessa**[1]

1 Department of Medical Laboratory Science, College Health Science, Debre Markos University, Debre Markos, Ethiopia, 2 Department of Midwifery, College Health Science, Debre Markos University, Debre Markos, Ethiopia

* gebredemeke@yahoo.com

## Abstract

### Background

Intestinal parasitosis is a common disease that causes misery and disability in poor populations. The number of individuals affected is staggering. From two billion peoples who harbor parasites worldwide, 300 million suffer severe morbidity and more than 25% of pregnant women are infected with hookworm, which causes intestinal bleeding and blood loss, and has been most commonly associated with anemia. Intestinal parasite infection during pregnancy has been associated with iron deficiency, maternal anemia, and impaired nutritional status, as well as decreased infant birth weight.

### Objective

This study aimed to assess the effects of intestinal parasite infection on hematological profiles of pregnant women attending antenatal care in Debre Markos Referral Hospital from December 2017 to February 2019.

### Method

A prospective cohort study design was conducted among 94 intestinal parasite-infected pregnant women as an exposed group and 187 pregnant women free from intestinal parasite were used as a control group.

The effect of intestinal parasites on hematological profiles of pregnant women was assessed at Debre Markos Referral Hospital antenatal care ward.

Socio-demographic data and nutrition status were assessed by using structured questionnaires and mid-upper arm circumference (MUAC), respectively. Two ml of venous blood and 2 gm of stool samples were collected to analyze the hematological profiles and detect

**Data Availability Statement:** All relevant data are within the paper and it Supporting Information files.

**Funding:** Fund received author: Gebreselassie Demeke Funder:- Debre Debre Markos University The funders had no role in study design, data collection and analysis, decision to publish, or preparation of the manuscript.

**Competing interests:** The authors have declared that no competing interests exist.

intestinal parasites, respectively. Wet mount and formol-ether concentration (FEC) techniques were used to detect intestinal parasites. Hematological profile was analyzed using Mind ray BC-3000 plus instrument. Data were double entered into EpiData version 3.1 software and exported to SPSS version 24 software for analysis. Results were presented using tables and graphs. Associations of hemoglobin levels with intestinal parasitic infections were determined using binary logistic regression models. $P \leq 0.05$ was considered statistically significant. The mean hematological profile difference between parasite-infected and parasite-free pregnant women was computed using independent $t$-test.

## Results

In the present study, the predominant parasites identified were *Entamoeba histolytica*, hookworm, *Giardia lamblia*, *Schistosoma mansoni*, and *Ascaris lumbricoides*. About 8.2% of intestinal parasite-infected pregnant women had mild anemia while 4% had moderate anemia. Only 1.2% of intestinal parasite-free pregnant women developed moderate anemia. The mean HGB, HCT, MCV, MCH, and MCHC values of intestinal parasite-infected pregnant women were 12.8g/dl, 38.2%, 94.7fl, 33.1pg and 34.7g/dl, respectively. But the mean HGB, HCT, MCV, MCH and MCHC values of pregnant women who were free from intestinal parasites were 14.4 g/dl, 39.8%, 94.9fl, 33.9pg and 35.5g/dl, respectively. Anemia was strongly associated with hookworm (AOR = 21.29, 95%CI: 8.28–54.75, *P<0.001*), *S.mansoni* (AOR = 63.73, 95% CI: 19.15–212, *P<0.001*) and *A.lumbricoide* (AOR = 14.12, 95% CI 3.28–60.65, *P<0.001*).

## Conclusion

Intestinal parasitic infection in pregnant women caused adverse impact on hematological profiles and was an independent predictor of anemia. Intestinal parasitic infection significantly decreased pregnant the level of HGB, HCT, MCV, MCH, and MCHC values. To minimize maternal anemia deworming could be good before pregnancy.

## Introduction

Intestinal parasites live in the gastrointestinal tracts of humans. Of these, soil-transmitted helminths (STHs) inflict a substantial burden of morbidity on poor populations living in tropical and subtropical regions [1].

Intestinal parasitosis is a common disease that continues to causing misery and disability among poor populations. From two billion peoples harboring parasites worldwide, 300 million suffer from severe morbidity. The World Health Organization (WHO) estimated that schistosomiasis and soil-transmitted helminthiasis represented more than 40% of the disease burden due to all tropical diseases, excluding malaria [2].

In the developing world, pregnant women, their infants, and children frequently experience undernutrition (macronutrient and micronutrient) and repeated infections, including parasitic infections, which lead to adverse consequences that can continue from generation to generation. Intestinal helminth infections alter different types of micronutrients in pregnant women and children [3].

Tropical diseases such as intestinal helminthiases, filariasis, and malaria have a high impact on reproductive health. Many cases of unexplained pregnancy loss are due to undiagnosed tropical diseases. Malnutrition or anemia caused by intestinal worms may be worsened during pregnancy and make the pregnancy difficult [4].

Anemia is defined as a condition in which there is less than the normal hemoglobin (HGB) level in the body, which decreases the oxygen-carrying capacity of red blood cells to tissues. Anemia during pregnancy is an important contributor to maternal mortality. Anemia can be classified as mild, moderate, and severe. Anemia is common in developing countries because of poor nutritional status and highly prevalent parasitic infestation [5].

Although anemia in pregnancy is multi-factorial, poor nutrition and infection are also common causes. In Sub-Saharan Africa, soil-transmitted helminths (STH) including hookworm, urogenital schistosomiasis, and other parasitic infections such as malaria contribute to the high anemia rates in women and young children [4,6].

Helminthic infection prevalence up to 50% has been documented in some regions in Sub-Saharan Africa [7]. More than 25% of pregnant women are infected with hookworm, which causes intestinal bleeding and blood loss, and has been most commonly associated with anemia [7,8]. Although there are many studies conducted in Ethiopia that have reported the magnitude of intestinal parasitic infections, there is a paucity of published data particularly on the effects of intestinal parasites on hematological profiles among pregnant women. Therefore, this study aimed to assess the effects of intestinal parasites on the hematological profiles of pregnant women. The results of the study could help concerned stakeholders to take action on the prevention of intestinal parasitic infection in pregnant women.

## Materials and methods

### Study area, design, and period

A prospective cohort study was conducted among pregnant women attending antenatal care at Debre Markos Referral Hospital, Ethiopia from December 2017 to February 2019. Debre Markos Referral Hospital is located in Debre Markos town. The town is located 300 km northwest of the capital Addis Ababa and 265 km southeast of the capital of Amhara National Regional State, Bahirdar. The climatic condition is generally humid, with a mean annual temperature of 16°C and rainfall of 1308 mm [9]. The dominant proportion of soil particles in Debre Markos is clay, which has clay content ranging from 50%– 73%, silt faction 15.88%–40.21%, and sand fraction 0.36% - 13.28%. The soil liquid limit range of Debre Markos is from 45%– 68% [10].

### Study population and sample size determination

Pregnant women attending the antenatal care unit at Debre Markos Referral Hospital fulfilling the inclusion criteria were consecutive conveniently enrolled in the study. The sample size was calculated using OpenEpi version 3.04.04 software considering two-sided significant level $\alpha$ = 0.05, power (1-beta) chance of detecting 80%, the proportion of disease in non-exposed = 0.1 ratio of sample size for non-exposed = 2 and proportion of disease in exposed group = 0.217.

$$p = p1 + rp2/r + 1, \ r = 1/2.$$

$$n1 = \frac{[Z\alpha/2\sqrt{(1+1/r)p1-p} + \beta\sqrt{p1(1-p1) + p2(1-p2)/r}]^2}{(P1-p2)^2}$$

The sample size of the exposed group was 103 and the non-exposed group was 206. The

total sample size was 309 [11]. Since 28 pregnant women withdrawn from participation, our study included 94 intestinal parasite-infected and 187 intestinal parasite-free pregnant women.

## Inclusion criteria

Those women at the first trimester who did not take the ant-parasite drug before two weeks, and had no chronic disease were considered as study participants.

## Exclusion criteria

Those who had known chronic disease, HIV patients, and have taken anti-helminthic drugs within the last two weeks.

## Data collection instrument and laboratory procedure

Before starting data collection each participant was informed about the objectives of the study and written informed consent was obtained from the study participants before data collection Socio-demographic data were collected using a structured questionnaire while laboratory data and other clinical data were collected at first, second and third trimester. Stool specimens were collected from each participant using a clean, leakproof, and sterile stool cup and processed using two parasitological techniques. After collection, three slides were prepared for each participant and a direct wet mount stool smear was microscopically examined by three medical parasitologists. Freshly voided stool specimen was directly examined microscopically within 15 minutes to identify intestinal parasite trophozoite. Direct wet mount stool examination was performed at Debre Markos Referral Hospital parasitology laboratory. The rest 1gm of stool was preserved with 10% formalin and transported to Debre Markos University for further examination using the formol-ether concentration method. In addition to a biological sample, anthropometric measurement was used to assess nutritional status. MUAC was measured to assess the nutritional status of pregnant women. The MUAC values below 23 cm were considered as an indicator of malnutrition for pregnant women [12].

**Formol-ether concentration method.** Stool specimen was processed following formol-ether concentration standard operation procedure. One gram of stool was added in a clean conical centrifuge tube containing 7ml, of 10% formol water. The suspension was filtered through a sieve into a 15ml conical centrifuge tube. Then 4 ml of diethyl ether was added to the formalin solution and the content was centrifuged at 300 rpm for 1 minute. The supernatant was discarded and stool smears were prepared on sterile slides from the sediment. Finally, the slides were examined under a microscope with a magnification power of 10x objective lenses to identify ova of helminths and 40x objectives for identification of trophozoite and cysts of intestinal protozoan [13].

**Complete blood cell counts.** Two ml of venous blood was collected using a sterile syringe with a needle and transferred to an EDTA tube for hematological profile analysis. Complete blood cell count was performed using Mind ray BC-3000 plus hematological analyzer.

Pregnant women hemoglobin level equal to or greater than 11g/dl was normal, 10–10.9g/dl is mild, 7–7.99g/dl is moderate, and < 7g/dl was considered as severe anemia[14].

## Statistical analysis

Data were double entered into EpiData version 3.1 software and exported to SPSS version 24 software for analysis. Data were presented using tables and graphs. The prevalence of intestinal parasites and anemia were described using frequency. The Association of hemoglobin levels with intestinal parasites was computed using logistic regression model. The association of

predictors with outcome variables was determined by using binary logistic regression analysis. $P \leq 0.05$ were considered statistically significant. Independent $t$-test was calculated to determine the mean difference of hematological profiles between intestinal parasites- infected pregnant women and intestinal parasite- free pregnant women.

## Data quality assurance

To assure the quality of data, before data collection, on-site training was given for data collectors on how to collect sociodemographic data and how to draw venous blood specimens. Reagents and instruments quality were checked for expiry dates and any functional problems. Site assessment and pre-test were done before data collection to optimize laboratory setup. This was done in samples obtained from 30 of the study participants. All data quality control tools were considered. Stool specimens were checked for their quantity whereas stool specimens contaminated with soil and urine were discarded and recollected. Lysed blood was discarded and recollected to optimize specimen quality. The stool specimens were examined in triplicate independently by three medical parasitologists. Data were checked for completeness before entry and analysis.

**Ethical consideration.**   Ethical clearance was obtained from the research and ethical review committee of the College of Health Sciences, Debre Markos University, (Ref; CHS/R/C/S/C/31/11/2017). All the information obtained from the study participants were coded to keep confidentiality. Clinically important results were communicated with clinicians for appropriate intervention.

## Results

### Socio-demographic characteristics

A total of 281 (94 intestinal parasite-infected and 187 non-infected) pregnant women at their first trimester were included in the study. Among them, 18 and 10 withdrew from participation at their second and third trimesters, respectively. Only 253 were followed till the end of the study. The study participants withdrew from the study was due to a change of residence facility, transferring to a nearby facility, and absent from follow-up due to unknown reasons. The majority of study participants were between the age ranges of 26–35 years. The mean age of intestinal parasite-infected pregnant women was 26 years (±3.95 SD) while it was 26 years (±4.01 SD) for intestinal parasite-free group. The majority of exposed and non-exposed study participants were from urban areas and more than 40% of the study participants attended college or above (Table 1).

### Dietary characteristics and nutritional status of pregnant women

From all the study participants 210 (74.7%) pregnant women did take iron folate during the first trimester.

Both exposed and non-exposed groups had similar nutritional habits. In this study, 220 (78.3%) of pregnant women took food three times within 24 hours. Seventy-nine (28.1%) of pregnant women feed meat, milk, or milk products once a month. Feeding behaviors of pregnant women vary from person to person. The participants had good habits of fruit and vegetable intakes. One hundred and twenty-eight (45.6%) of pregnant women feed fruit and vegetables at least once within 24 hours (Table 2).

In addition to the daily food intake habits of pregnant women; MUAC was assessed to estimate nutritional status. The majority of the study participants (79%) had MUAC >23 cm

**Table 1. Socio-demographic characteristics of study participants attending antenatal care at Debre Markos Referral Hospital from December 2017 to February 2019.**

| Character | Study group | | Control group | |
|---|---|---|---|---|
| | Frequency | Percentage | Frequency | Percentage |
| **Age category** | | | | |
| **18–25** | 46 | 48.9 | 71 | 38 |
| **26–35** | 48 | 51.1 | 116 | 62 |
| **Residence** | | | | |
| Rural | 16 | 17 | 16 | 8.6 |
| Urban | 78 | 83 | 171 | 91.4 |
| Occupation | | | | |
| Government employer | 27 | 28.7 | 57 | 30.5 |
| House wife | 33 | 35.1 | 48 | 25.6 |
| Merchant | 25 | 26.6 | 31 | 16.6.4 |
| Others | 9 | 9.6 | 51 | 27.3 |
| Educational status | | | | |
| Not read and write | 15 | 16 | 14 | 7.5 |
| Primary | 10 | 10.6 | 16 | 8.6 |
| Secondary | 25 | 26.6 | 76 | 40.6 |
| College and above | 44 | 46.8 | 81 | 43.3 |

indicating good nutritional status. The mean, median and interquartile range of MUAC was 25.7 cm, 25 cm, and 27.75±7.75 cm, respectively (Table 3).

## Prevalence of intestinal parasites among pregnant women

Among 281 first-trimester pregnant women, 94(33.5%) were infected by at least one intestinal parasite. Among detected intestinal parasites hookworm dominated prevalence with a rate of

**Table 2. Dietary characteristics of pregnant women attending antenatal care at Debre Markos Referral Hospital, Ethiopia.**

| Food intake frequency within 24 hours | Frequency | Percent |
|---|---|---|
| Two times | 37 | 13.2 |
| Three times | 220 | 78.3 |
| Four times | 24 | 8.5 |
| Total | 281 | 100 |
| Meat, milk, and milk product intake habit | | |
| Daily | 21 | 7.5 |
| Two times within a week | 67 | 23.8 |
| One time within a week | 66 | 23.5 |
| One time within a month | 79 | 28.1 |
| One time more than a month | 48 | 17.1 |
| Total | 281 | 100 |
| Fruit and vegetable intake habit | | |
| Daily | 128 | 45.6 |
| Two times within a week | 98 | 34.9 |
| One time within a week | 48 | 17.1 |
| One time within a month | 1 | 0.4 |
| One time more than a month | 6 | 2.1 |
| Total | 281 | 100 |

**Table 3. The MUAC feature of pregnant women attending ANC at Debre Markos Referral Hospital at different trimesters.**

| Character | Mean | Media | Maximum | Minimum |
|---|---|---|---|---|
| MUAC | | | | |
| First trimester | 25.5cm | 25cm | 33cm | 20cm |
| Second trimester | 25.9cm | 25cm | 33cm | 20cm |
| Third trimester | 25.7cm | 25.5cm | 35.5cm | 21cm |

48(17.1%), *Entamoeba histolytica* 15(5.03%), *Schistosoma mansoni* 9 (3.2% and *Giardia lamblia* 5(1.6%) were also among the reported intestinal parasites. The prevalence of intestinal parasites decreased to 33.1% at the second trimester of pregnancy. Except for *E.histolytica* and *Ascaris lumbricoides*, the prevalence of all other intestinal parasites decreased. Unfortunately, new infections were detected during the third visit in both control and exposed groups.

The overall prevalence of intestinal parasites among pregnant women in their third trimester was increased to 38.3%. Prevalence with the respective prevalence of hookworm, *S.mansoni*, and Taenia species 13.4%, 5.1% and 4% (Fig 1).

## Effects of intestinal parasite on hematological profiles

In this study, 5% (95% CI: 3.5–6.8) of pregnant women were anemic with a hemoglobin level below 11 g/dl. During follow-up from the first trimester to the third trimester, the prevalence of anemia among intestinal parasite-infected pregnant women was 12.2%. Of which mild and moderate anemia accounts for 8.2% and 4%, respectively. The prevalence of anemia among intestinal parasite-free pregnant women was 1.2% and the level of anemia was mild. The mean hemoglobin level of intestinal parasite-infected and parasite-free pregnant women was 12.8 g/dl and 14.4 g/dl, respectively. Anemia was significantly associated with intestinal parasites ($P$ = 0.001). Intestinal parasite-infected pregnant women had higher odds of developing anemia than intestinal parasite-free (AOR = 2.5; 95% CI = 1.683–3.807). Comparatively, the mean HGB, MCV, HCT, MCH, and MCHC levels of intestinal parasite-infected pregnant women were lower than intestinal parasite-free pregnant women. The difference was observed in the mean hematological profiles between intestinal parasite-infected and intestinal parasite-free pregnant women. Whereas, total white blood cells (WBCs), Lymphocyte (LYM), Granulocyte (GRA), and Platelets (PLTs) increased in pregnant women infected by intestinal parasites. Based on independent *t*-test result, the mean difference of hematological profiles was significantly associated with intestinal parasites among pregnant women (Table 4).

The magnitude of mean hemoglobin difference between intestinal parasite-free and intestinal parasite-infected pregnant women was 1.54g/dl. There was a strong association between anemia and intestinal parasites. Pregnant women infected with hookworm (AOR = 21.29, 95% CI: 8.28–54.75, *P<0.001*), *Schistosoma mansoni* (AOR = 63.73, 95% CI: 19.15–212, *P<0.001*) and *Ascaris lumbricoides* (AOR = 14.12, 95% CI: 3.28–60.65, *P<0.001*) had higher odds of being anemic than non-infected pregnant women.

The overall prevalence of anemia among pregnant women at their first ANC was 21/281 (7.5%). The prevalence of anemia among intestinal parasite-infected participants was 13/94 (17%). Among these, 13/94 (13.8%) and 3/94 (3.2%) intestinal parasite-infected pregnant women developed mild and moderate anemia, respectively. Only 5/187 (2.7%) of parasite-free pregnant women were developed mild anemia. Anemia was strongly associated with intestinal parasitic infection with (*P<0.001*). Intestinal parasite-infected pregnant women were 7.47 (AOR = 7.47, 95% CI: 1.058–3.461) times more likely to developed anemia than intestinal parasite-free pregnant women during the first trimester. The prevalence of anemia among

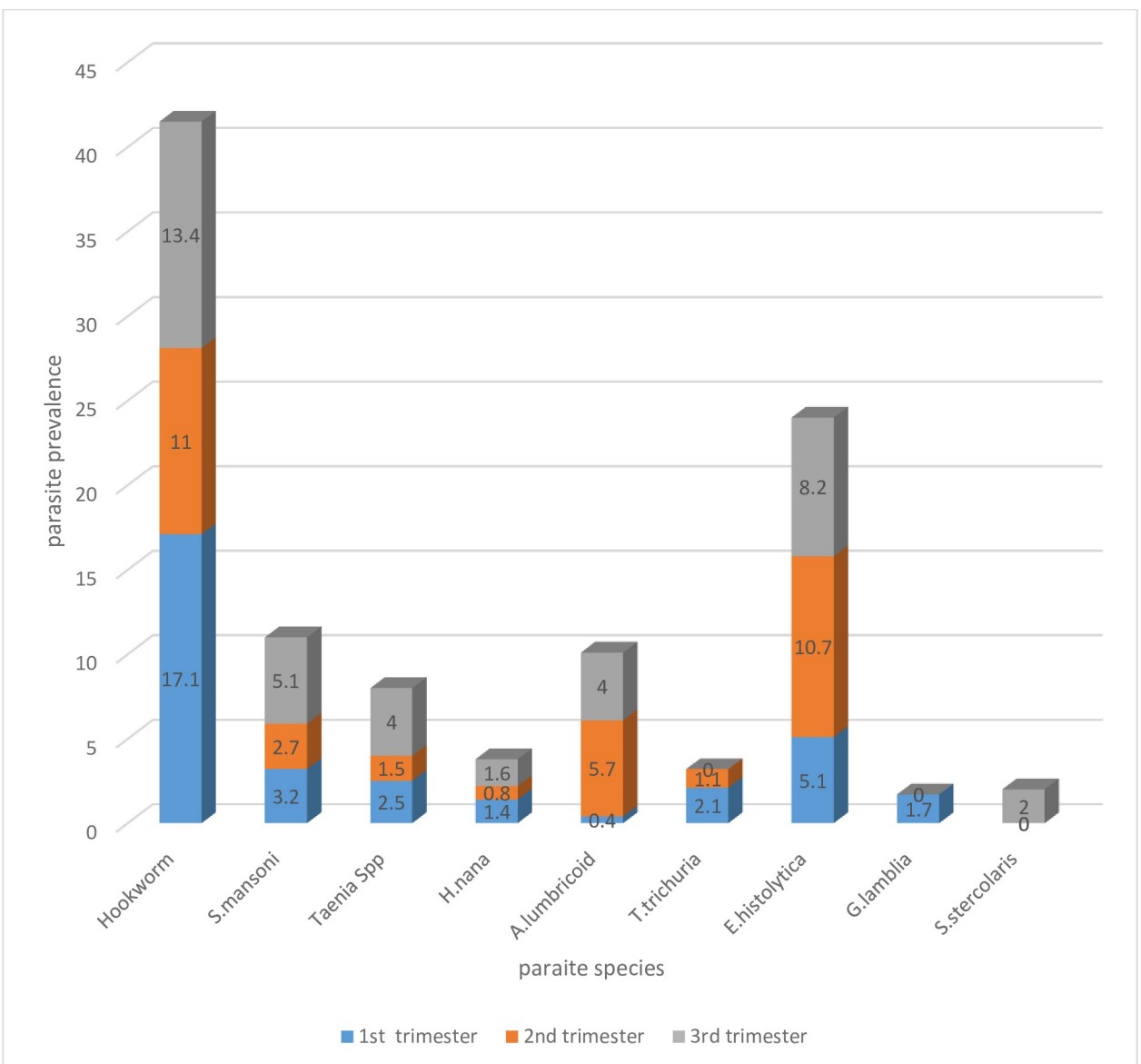

**Fig 1. The distribution of intestinal parasites stratified by trimester among pregnant women at Debre Markos Referral Hospital, Ethiopia.**

pregnant women of the second trimester was 13/263 (4.9%). Among intestinal parasite-infected second trimester pregnant women, 12/87 (13.8%) of them were anemic. From these, mild and moderate anemia account for 8/87(9.2) and 4/87(4.6%), respectively. Only one person was anemic among intestinal parasite-free pregnant women. There was a significant difference in the mean hematological profiles between intestinal parasite-infected and non-infected pregnant women. Intestinal parasite-infected pregnant women were HGB = 12.8g/dl, HCT = 37.9%, MCV = 94.9fl, MCH = 33.7pg, MCHC = 35.3g/dl and t HGB = 14.8g/dl, HCT = 40.5%, MCV = 95.1fl, MCH = 33.6 pg, and MCHC = 35.5g/dl for intestinal parasite-free pregnant women.

The prevalence of anemia among intestinal parasite-infected third-trimester pregnant women was 6/91(6.2%). In the third trimester, none of intestinal parasite-free pregnant

**Table 4. The mean hematological profile value difference between intestinal parasite-infected and intestinal parasite-free pregnant women at Debre Markos Referral Hospital, Ethiopia.**

| hematological parameter (unit) | Intestinal parasite infection status | | Mean difference | t-test | P-value | 95% CI |
|---|---|---|---|---|---|---|
| | Yes | No | | | | |
| HBG (/ g/dl) | 12.8234 | 14.3643 | 1.54 | 6.98 | < 0.001 | 1.177–1.905 |
| HCT (%) | 38.1781 | 39.7869 | 1.61 | 5.04 | < 0.001 | 0.974–2.235 |
| MCV(fl) | 94.7464 | 94.8732 | 0.126 | 0.26 | 0.792 | - 0.817–1.070 |
| MCH (pg) | 33.1032 | 33.8728 | 0.77 | 4.22 | < 0.001 | 0.371–1.168 |
| MCHC(pg/dl) | 34.7061 | 35.5041 | 0.79 | 4.84 | < 0.001 | 0.468–1.128 |
| RBC(/ul) | $4.1 \times 10^6$ | $4.3 \times 10^6$ | $0.2 \times 10^6$ | 3.6 | <0.001 | 0.07–0.22 |
| RDW-SD(fl) | 45.1 | 46.6 | 1.5 | 1.12 | 0.264 | -1.11–4.06 |
| RDW-CV (%) | 13.2 | 13.3 | 0.1 | 0.11 | 0.906 | -0.155–0.175 |
| WBC(/ul) | $8.4 \times 10^3$ | $8.3 \times 10^3$ | -0.1 | -0.65 | 0.513 | -0.591–0.315 |
| LYM (%) | 2.47 | 2.08 | -0.39 | -2.2 | 0.026 | -0.720–0.047 |
| MID (%) | 0.82 | 0.91 | 0.09 | 0.99 | 0.323 | 0.094–0.284 |
| GRAN (%) | 6.89 | 6.71 | -0.18 | -0.37 | 0.711 | -1.098–0.749 |
| PLT(/ul) | $224.9 \times 10^9$ | $218.5 \times 10^9$ | -6.4 | -1.28 | 0.199 | -16-2-3.4 |

HGB: Hemoglobin, HCT = Hematocrit, MCV: Mean Corpuscular Volume, MCH: Mean Corpuscular Hemoglobin, MCHC: Mean Corpuscular Hemoglobin Concentration, RDW-SD: Red Blood Cell Distribution Wide Standard Deviation, RDW-CV: Red Blood Cell distribution width-Variation Coefficient, LYM: Lymphocyte, MID: MID-range absolute count, GRAN: Granulocytes, PLT: Platelet.

women developed anemia. Pregnant women at their third trimester had better hemoglobin concentration and other hematological profiles compared to their profiles at the first and second trimesters.

## Discussion

Several factors cause anemia during pregnancy. Among these intestinal parasitic infection is the most common one. In Ethiopia, intestinal parasites are common which have negative effects on mothers' hemoglobin concentration and infant weight. Therefore, the purpose of this study was to assess the impact of intestinal parasites on hematological profiles of pregnant women attending antenatal care at Debre Markos Referral Hospital.

Even though there were several studies conducted in Ethiopia to show the prevalence and risk factors of intestinal parasites but only a few studies were examined to show the burden of intestinal parasites in pregnant women. Intestinal parasites have adverse effects on hematological profiles of pregnant women. In the current study, the mean age of the study participants was 26. It is similar to other studies conducted in Ethiopia [15]. The nutritional status of pregnant women was assessed using Mid Upper Arm Circumference (MUAC) measurement. Seventy-nine percent (79%) of the study participants MUAC was greater than 23 cm. which indicates good nutritional status. The mean MUAC value of pregnant women was 27.75 ±7.75cm. It is higher than other studies conducted at Wondo Genet district in Ethiopia which was 22.8 ±1.9 cm [15]. Women whose MUAC greater than 23cm were less likely to develop anemia due to nutritional deficiency [16]. Pregnant women whose MUAC below 23cm were considered as an indicator of malnutrition status [17]. The nutritional difference between the current finding and other studies might be due to population differences and socio-demographic characteristics. In Ethiopia, there are many cultures which determine the feeding habits of the community and in our study, most of the study participants were from urban area which had better life quality. The overall prevalence of intestinal parasites among pregnant

women attending antenatal care (ANC) at Debre Markos Referral Hospital was 34.9% (95% CI: 31.7% -38.2%,) which is in line with other study conducted in Ethiopia [18,19]. But this finding is lower than other studies conducted in Indonesia and Venezuela with the prevalence of intestinal parasites among pregnant women was 69.7% and 73.9%, respectively [20,21], and higher than other studies conducted in Uganda. The prevalence of intestinal parasites among pregnant women was (16%) [22]. The difference might be seasonal variation, rainfall, soil types, and lifestyle differences of the study population. Hookworm prevalence was (11%) which was higher than other intestinal parasites. Hookworm is one of the most commonly known intestinal parasites that cause anemia. Similar studies conducted in Ethiopia and Uganda with the prevalence of hookworm were similar to our finding [19,23].

Determination of hemoglobin concentration was one of the parameters which were important to assess anemia. In our study, the prevalence of anemia among both intestinal parasite-infected and intestinal parasite-free pregnant women was 5%. The prevalence of anemia among intestinal parasite-infected first, second and third trimester pregnant women was 17%, 13.8% and 6.2%, respectively. But the prevalence of anemia among intestinal parasite-free pregnant women at first, second, and third trimester was 2.7%, 0.6% and 0.0%, respectively. The difference in anemia prevalence between intestinal parasite-infected and parasite-free pregnant women was a direct indicator for parasite effects on anemia. This finding is lower than other studies conducted among Sudan's refuge pregnant women, community-based study in western Ethiopia, and hospital-based study in southern Ethiopia with the prevalence of anemia was 32.2%, 17.5%, and 31.5%, respectively [24–26]. Due to parasite mode of transitions and less sanitation practice in refugee camps increased intestinal parasite prevalence and their effects. Another reason also may be due to study area, population, and lifestyle differences.

In our finding anemia was strongly associated with intestinal parasitic- infection ($P<0.001$). This finding is similar to a study conducted in the southern part of Ethiopia ($P<0.001$). Parasite-infected pregnant women were 6.97 more likely to develop anemia than parasite-free ones. The mean hemoglobin difference between intestinal parasite-free pregnant women and parasite-infected pregnant women was 1.54(95% CI: 1.107–1.974). This was in line with similar studies conducted in different parts of Ethiopia [15,16]. Among many factors that cause anemia; intestinal parasite infection is one factor. In this study hookworm ($P<0.001$), *Schistosoma mansoni* ($P<0.001$) and *Ascaris lumbricoid* ($P<0.001$) were strongly associated with anemia. This finding was similar with other studies conducted in Ethiopia [25] and Venezuela [4].

The mean hematological profile of pregnant women were (HGB = 13.8gm/dl, HCT = 39.2%, MCV = 94.8fl, MCH = 33.6pg, and MCHC = 35.2 g/dl). This finding is comparable with the study conducted at Debre Berhan Referral Hospital of Ethiopia. The hematological profile were (HGB = 13.8gm/dl, HCT = 39.2%, MCV = 94.8fl, MCH = 33.6pg, and MCHC = 35.2 g/dl) [27].

The similarity may be due to similar target groups, populations, and altitudes.

The mean hematological profiles difference were observed between intestinal parasite-infected and intestinal parasite-free pregnant women. The mean HGB and RBC indices value of intestinal parasite-infected were HGB = 12.8g/dl, HCT = 37.9%, MCV = 94.9fl, MCH = 33.7pg, MCHC = 35.3g/dl. But intestinal parasite free pregnant women hematological profiles were (HGB = 14.8g/dl, HCT = 40.5%, MCV = 95.1fl, MCH = 33.6 pg, and MCHC = 35.5g/dl). This study is comparable with other studies conducted on pregnant women [15,27]. Therefore, parasitic infections had adverse impacts on the hematological profiles of pregnant women. In our study hematological profiles such as total white blood cells (WBCs), lymphocyte (LYM), granulocyte (GRA) and platelets (PLTs) were increased. In most

of the infections, total WBCs are increased to fight against the foreign organism. In helminthic infections, it is common for the increment of eosinophils while in protozoal infections WBC has increased. In general, the increments of the above-mentioned hematological profiles were due to infections.

Previous studies reported the effects of intestinal parasites on hemoglobin levels and red blood cell indices; however, the effects of intestinal parasites on the hematological profiles of pregnant women were not yet well studied. Our study assessed the effects of intestinal parasites on the hematological profiles in pregnant women at Debre Markos Referral hospital. The current study has limitations as it didn't use sensitive techniques to detect intestinal parasites which might underestimate the prevalence of intestinal parasites. Besides, the effect of intestinal parasites on the hematological profiles was not assessed in the fetus and even after delivery. Besides these limitations, we believe that the results of this study can contribute to the prevention of intestinal parasites and minimizing their impacts during pregnancy. Future studies should use sensitive techniques and investigate the effects of intestinal parasites on the hematological profiles in the fetus and infants.

## Conclusion

Intestinal parasitic infection in pregnant women was common and had adverse impacts on hematological profiles. Some hematological profiles of intestinal parasite-infected pregnant women were decreased which leads to anemia in pregnant women. But some of the hematological profiles such as total white blood cells (WBCs), lymphocyte (LYM), granulocyte (GRAN) and platelets (PLTs) were increased. As to minimize maternal anemia during pregnancy deworming before pregnancy and personal hygiene and environmental hygiene may be important.

## Supporting information

**S1 Data.**
(SAV)

## Acknowledgments

We would like to express our deepest gratitude to Debre Markos University for financial input, materials, and other valuable logistic supports that were extremely important for the achievement of this research. Our thanks also extend to all members of Debre Markos University, Medical Laboratory Science staffs, especially their intellectual inputs. Finally, we would like to thank all study participants.

## Author Contributions

**Conceptualization:** Gebreselassie Demeke, Getachew Mengistu, Aster Shiferaw, Hylemariam Mihiretie Mengist, Tebelay Dilnessa.

**Data curation:** Gebreselassie Demeke, Abtie Abebaw, Aster Shiferaw, Hylemariam Mihiretie Mengist.

**Formal analysis:** Gebreselassie Demeke, Abtie Abebaw, Milkiyas Toru, Molla Yigzaw, Hylemariam Mihiretie Mengist, Tebelay Dilnessa.

**Funding acquisition:** Gebreselassie Demeke.

**Investigation:** Gebreselassie Demeke.

**Methodology:** Gebreselassie Demeke, Getachew Mengistu, Abtie Abebaw, Milkiyas Toru.

**Project administration:** Gebreselassie Demeke.

**Resources:** Gebreselassie Demeke.

**Software:** Gebreselassie Demeke, Getachew Mengistu, Molla Yigzaw, Hylemariam Mihiretie Mengist, Tebelay Dilnessa.

**Supervision:** Gebreselassie Demeke, Hylemariam Mihiretie Mengist.

**Validation:** Gebreselassie Demeke, Milkiyas Toru.

**Visualization:** Gebreselassie Demeke.

**Writing – original draft:** Gebreselassie Demeke, Abtie Abebaw, Milkiyas Toru, Molla Yigzaw, Aster Shiferaw, Hylemariam Mihiretie Mengist, Tebelay Dilnessa.

**Writing – review & editing:** Gebreselassie Demeke, Getachew Mengistu, Abtie Abebaw, Milkiyas Toru, Molla Yigzaw, Aster Shiferaw, Hylemariam Mihiretie Mengist, Tebelay Dilnessa.

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
