## [Decision Letter · Decision Letter 0]

30 Dec 2020

PONE-D-20-36685

Effects of intestinal parasite infection on hematological profile of Pregnant Women Attending Antenatal care in Debre Markos Referral Hospital, North West Ethiopia. Institution based Prospective cohort study

PLOS ONE

Dear Dr. Demeke,

Thank you for submitting your manuscript to PLOS ONE. After careful consideration, we feel that it has merit but does not fully meet PLOS ONE’s publication criteria as it currently stands. Therefore, we invite you to submit a revised version of the manuscript that addresses the points raised during the review process.

Experts in the field handled your manuscript, and we are grateful for their time and contributions. Although some interest was found in your study, numerous major concerns arose that require your attention. Your manuscript received many comments, which I feel if addressed properly, will make this study stronger and contribute to the research topic. Please respond to ALL of the reviewers' comments in your revised manuscript.

We look forward to receiving your revised manuscript.

Kind regards,

Frank T. Spradley

Academic Editor

PLOS ONE

4. Please ensure you have thoroughly discussed any potential limitations of this study within the Discussion section, including the potential impact of confounding factors.

5. We note you have included a table to which you do not refer in the text of your manuscript. Please ensure that you refer to Table 4 in your text; if accepted, production will need this reference to link the reader to the Table.

6. We noticed you have some minor occurrence of overlapping text with the following previous publications, which needs to be addressed:

- https://www.ncbi.nlm.nih.gov/pmc/articles/PMC1522064/

- https://www.longdom.org/open-access/prevalence-of-iron-deficiency-anemia-and-determinants-among-pregnantwomen-attending-antenatal-care-at-woldia-hospital-ethiopia-2161-0509-1000201.pdf

- https://journals.plos.org/plosntds/article?id=10.1371%2Fjournal.pntd.0002724

The text that needs to be addressed involves, primarily, the Introduction section of your manuscript. In your revision ensure you cite all your sources (including your own works), and quote or rephrase any duplicated text outside the methods section. Further consideration is dependent on these concerns being addressed.

Reviewers' comments:

Reviewer's Responses to Questions

**Comments to the Author**

1. Is the manuscript technically sound, and do the data support the conclusions?

Reviewer #1: Partly

Reviewer #2: No

Reviewer #3: No

Reviewer #4: Partly

Reviewer #5: Yes

Reviewer #6: Partly

Reviewer #7: No

2. Has the statistical analysis been performed appropriately and rigorously? 

Reviewer #1: Yes

Reviewer #2: Yes

Reviewer #3: No

Reviewer #4: No

Reviewer #5: Yes

Reviewer #6: Yes

Reviewer #7: N/A

3. Have the authors made all data underlying the findings in their manuscript fully available?

Reviewer #1: No

Reviewer #2: No

Reviewer #3: Yes

Reviewer #4: Yes

Reviewer #5: No

Reviewer #6: No

Reviewer #7: Yes

4. Is the manuscript presented in an intelligible fashion and written in standard English?

Reviewer #1: Yes

Reviewer #2: No

Reviewer #3: No

Reviewer #4: Yes

Reviewer #5: No

Reviewer #6: No

Reviewer #7: No

5. Review Comments to the Author

Reviewer #1: This study is about the effects of intestinal parasite infection on hematological profile of Pregnant Women Attending Antenatal care in Debre Markos Referral Hospital from December 2017 to February 2019. The manuscript needs major revision.

In abstract (lines 27-29) and in introduction (lines 90-92); “More than 25% of pregnant women are infected with hookworm, which causes intestinal bleeding and blood loss, and has been most commonly associated with anemia.” According to the reference that the authors refer to it this results is related to Allada in Benin, but the authors wrote it as a general rule, so it has to be revised.

The hematological factors studied in this manuscript give all intestinal parasitic infections the same value. While some of these infections, such as hookworms, may cause more anemia. Therefore, it is necessary to evaluate each parasitic infection separately.

The authors evaluated a lot of nutritional factors but the relation between them and parasitic disease is unknown.

Are the nutrition status for infected and control group same or not?

The pregnant women separated to 3 groups (first, second and third semester) but the results were presented in total.

The control group need to be matched with infected group (Socio- demographic characteristics and dietary characteristics and nutritional status).

The results of Socio- demographic characteristics and dietary characteristics and nutritional status should be expressed in relation to parasitic infection. While in this study these results generally stated.

The number of ethical statement needs to added.

Reviewer #2: Reviewer comments

Generally the manuscript is important to address effects of IPs among pregnant women; however, it needs major revision.

1. Abstract

Major comments

Objective of the study is still a proposal! It does not clearly indicate what objectives were addressed.

The method part is not clear!!

How many pregnant women were included both in the study and control groups?

The data collection (socio-demographic and lab based) should be included!!

How stool and blood samples were collected and processed?

What type methods do you used to detect the parasites and anemia parameters?

Method of data analysis is not clearly stated!! At what significant value did you accept the association? It should be included in the abstract data analysis part!!

In the result part, the prevalence of IPs should be included since they are the major predisposing factor in pregnant anemia! And also their effect in single and double or triple infections!! Is the anemia mild, moderate or severe both in the infected and control groups? In what percent?

The conclusion part should be produced based on your findings!! First what was the prevalence of IPs among pregnant women? Second which parameters of anemia are decreasing due to parasitic infection? Is the anemia mild, moderate or severe?

Poor recommendation? Are anthelminthic drugs safe for pregnant women? I doubt!!

Generally the abstract needs major revision!!

Minor comments:

line no 36, December/ 2017 to February 2019 should be corrected as “December 2017 to February 2019”

Revise the grammar!! P-value should be capital on line 48

2. Introduction

Major comments

Line no 75-77: in the first place the data is very old and also the reference you put [2] is not a WHO report rather “Rodriguez-Morales, A.J., et al, 2006” data in the reference list. So correction is needed!!

Basic information on pregnant associated anemia is not indicated! For instance,

What are the possible factors of anemia? It might be caused by physiologically or pathologically during pregnancy. However, you document does not clearly show anemia caused by physiological conditions based on the trimesters.

How do you know whether anemia is caused by physiological, malnutrition or parasitic infection??

At what parasitic load the parasite can cause anemia (is that in the light, moderate or heavy)? You have to be quite shore to this condition!! Most of the time anemia may be caused by the parasite if and only if the load is high.

In this study, nutrition and physiological condition of the pregnant are cofounding effects that might lead to false conclusion.

The main objective of the study is not clearly stated in the introduction part!! For instance,

Why you want to study in Debre Markos referral hospital? It might be good if you conduct in rural set ups?

Do you have any institution based information at Debre Markose referral hospital on the level of anemia and prevalence of IPs among pregnant women previously that attract you to conduct this research?

These issues listed above are the major limitation of the introductions parts of your study

Minor comments

The grammar should be seen with native speaker

Please do not repeat the whole word and Abbreviations! E.g Soil-transmitted 71 helminths (STHs) line no 71 and 86

3. Methods and materials

Major comments

The study area should be full described based on the parameters important for the existence of parasites in the area (altitude, soil-type, Temperature and rainfall.

As you mention in the (line no 98-99), you recruited pregnant women who fulfill your inclusion criteria. However, nothing is stated about the inclusion criteria. What was you inclusion criteria?

Once the pregnant fulfill the inclusion criteria, how do you select the pregnant women that participated in your study? It is vague whether you select them randomly or conveniently? The sampling frame is generally unclear!!

As you mention in the (line no 101-102), to calculate the sample size, you use the prevalence of anemia and among non-helminthic infected (10%) and helminthic infected pregnant women (21.7%), respectively. Where did you get this prevalence? You must indicate the source or reference!!!

In the exclusion criteria (line no 105-6), those taking anthelminthic drugs within the last 2 weeks were excluded. In the first place, did pregnant women take anthelminthic drug? At what stage? Secondly, why did you exclude only two weeks? In the re-infection, the parasite cycle needs to be detected by microscopy after 4 weeks. I am not clear about it!! I need your justification or you should revise based on scientific evidences!!

How many pregnant women did you select? How did you follow them? When did the lab samples collected? Is that during the 1st, 2nd and 3rd trimester? What did you do for those positive in the 1st trimester? Did you give them treatment and again follow in the 3rd? Completely unclear!!!!! Major limitation!

Why did you use direct microscopy and FECT methods together?? FECT might be enough since it is more sensitive than direct saline microscopy!!

The formol ether concentration test (FECT) should be revised critically based on the standard operating procedure!! As far as my knowledge is concerned, half gram of stool is enough to FECT, it is mixed with 10 ml of normal saline and sieved to test tube containing 7 ml of formalin and finally 3 ml of ether is added and the final volume is 10 ml. the revolution per minute is also is not correct!!

What parasites did you see with 10x objectives? How did you differentiate the cyst stages of amoeba (you mean by 10X)? Very unusual!!! Totally re-write the FECT principle and procedure clearly!!

As you mention in (line no 129), anthropometric measurement was used to assess the nutritional status. What was your guide line and at what point the pregnant women is considered as malnourished? Needs reference!!

How did you calculate the prevalence of IP infections and hemoglobin? Descriptive statistic should be included in the analysis part!!

Considering the ethical issue, what did you do for those pregnant women infected with IPs and those had low hemoglobin level? Leave the participants untreated is unethical!

Generally the method part should be critically modified and needs major revision by considering the study and control groups!!!!

Minor comments:

line no 93, Methods and materials should be substitute by “Materials and Methods”

line no 96, December/ 2017 to February 2019 should be corrected as “December 2017 to February 2019”

line no 105, anti-helminthic should be corrected as “anthelminthic”

line no 109, Socio demographic should be corrected as “socio-demographic”

line no 117, 1gm should be corrected as “One gram”

line no 135, p-value should be corrected as “P-value”

4. Results

Major comments

Clearly indicate the socio-demographic characteristics of the study and control groups since it is a cohort study!!

Table 1 needs revision! In my knowledge, frequency could not be expressed in ratio rather in number. For instance, merchant and others under the occupation are expressed in ratio.

From line no 163-168, your result showed that the majority of the pregnant women took foods that increase the iron level that also increases the hemoglobin level. My worry is the amount of each food item and the amount iron that they get from each food items is unclear. It is not a matter of eating two, three or four times rather the amount!! This limits your study result!!!

In my view, it could have been good if you assess the impacts parasitic infections on hemoglobin level from those pregnant who have MUAC >23 cm since it is possible to remove the cofounding effect of malnutrition (nutrition as a cofounder). What will be your response??

From line 182-195, you reported 33.5%, 33.1% and 38.5% IPs prevalence among pregnant women in the 1st, 2nd and 3rd trimester. I am no clear what do you do during the follow up. Once you get positive in the 1st trimester, you treat them and re-exam in the 2nd and the same thing in the 3rd? Then what did you do for those positive in the 1st trimester? Have treat them and follow them again in the 2nd and 3rd ? This is not clear! You must clearly put the procedure that you follow during the follow up in the method section otherwise; it is very difficult to understand this part of your result.

The prevalence of intestinal parasite among pregnant women both in the control and study groups across their trimesters is not clearly described in the result section. Major limitation!!

Prevalence of intestinal parasite among pregnant women (both the control and cases) across their trimester might be good if you put in table!

In table 4, it will be very good if you put the normal value in one column so that any reader can easily understand!!

From line no 207-210, the odds ratio gap is huge. What could be the possible justification? And also table 5 does not indicate the odds ratio of IPs infections vs anemia! I hope one table is missed that show the association of IPs with hemoglobin!!

The severity of anemia (normal, mild, moderate and severe) should be fully indicated both the study and the control groups!!!!!

Minor comments

Line no 151, I am not clear about 253. 58.4%.!! Needs revision specially the percentage!!

Line no 159 and 160, anti-helminthic should be corrected as “anthelminthic”

Line no 164, 220(78.3%) should be corrected as “220 (78.3%)”. This should be done throughout the document.

Line number 167, correct as 128 (45.6%) and numbers could bot ne written in the beginning of a sentences

Line number 202, (Table 3) should be corrected as (Table 4)

The grammar should be modified!

5. Discussion

Major comments

Line no 228-232, the first paragraph of your discussion is your statement of the problem why you want to conduct this research. It is not placed in its appropriate place. Better to remove it in the discussion part!!

In the discussion part, you must discuss your major results in relation to the others study. Needs modification and major revision throughout discussion part!!

Minor comments

Line no 252, Hook worm should be corrected as “Hookworm”

Line no 262,263,267,268, p<0.001 should be corrected as “P<0.001”

Line no 284, infect ted should be corrected as “infected”

6. Conclusion

Generally, the conclusion should be drown based on the main findings of your study (pregnant vs parasite, Parasite vs anemia, associated factors with pregnant anemia).

The recommendation should be specific to the study paricipants, the community and stakeholders.

Generally, these types of studies are important to prevent the burdens of IPs infections among pregnant women in endemic countries since they need strong attention. However, the objectives and the methods of the study should be clearly stated to get a valuable result and to reach in conclusion. It needs major revision and acceptance of it will be secured after the above stated comments are addressed.

Reviewer #3: The reviewer of this manuscript would like to appreciate authors for the original work. However, there are issues that require further revision and significant amendment before its further consideration for publication.

1. Topic: Seems good. But there is no/lack of link between the topic and major findings of the study. The topic talks about effect of intestinal parasites on the hematological profile of pregnant women. Whereas, in the result section only data on RBC indices are indicated. While hematological profile refers to all blood cells (including WBC indices) and platelets, which is missing in the document. Rather the study focuses on effect of intestinal parasite on anemia in pregnant women...

2. Abstract,

- No justification/rationale, gap to be filled by this work is not indicated

- Emphasis has given to data analysis, better to briefly describe the laboratory procedure aspects

- Conclusion doesn't supported by the finding, as there is no data on WBCs indices and platelets

3. Introduction,

Further refining of available literatures (recent) is important

No justification/rationale, gap to be filled by this work is not indicated

4. Method,

Sample size is small and difficult to draw conclusion from this, appropriate data collection procedures

was used., but data analysis seems shallow and weak to show effect of the infections on RBC indices

5. Discussion (require re-writing)

Very shallow and fragmentation/duplication of ideas/contents here and there

There is a mix of sections, methods and result parts are in the discussion session (this need further revision and re-

writing)

Lack of proper scientific reasoning/explanation/justification. e.g. in all cases explanation given to the differences observed in

the study from other available literatures are; differences in study area, population and life style of the pregnant women.

It could be possible in some case, but can't be in all case. Proper scientific explanation/interpretation is important

Reviewer #4: The manuscript has very interesting findings that the authors can analyze and show much better so improve the impact of this publication. Please find in the attachment my comments for each chapter of the manuscript.

Reviewer #5: Demeke et al. evaluate the impact of intestinal parasite infection on hematological profile of pregnant women attending ANC in Debre Markos Referral Hospital. The study is simple but the subject studied is of public health significance in Ethiopia and could inform policy makers in the efforts to combat intestinal parasitic infection and anemia in pregnancy in the region. While the study seems to have been properly conducted, some issues need to be addressed by the authors.

Major issues

Introduction:

It will benefit readers if the authors could provide a more current WHO statistic on the prevalence and burden of intestinal parasitic infections in Africa and Ethiopia.

The introduction also appears to lack information and previous reports on the impact of intestinal parasitic infection on hematological parameters.

Methods:

How was the sampling done? Was this random or systematic? A brief description should be provided.

The exclusion criteria states that patients with known chronic disease and HIV were not included. Was this information based on word of mouth from the patients or data was retrieved from the hospitals archive. This should be clarified.

Age is also a factor that have impact on the hematological profile. Adolescent girls are known to be at high risk for anemia and so are older women. Was there age limit to the study?

How many participants were excluded based on the exclusion criteria? A flow chart showing this information may be beneficial to the readers. The authors need to clarify the inclusion/exclusion criteria. Several underling conditions affect the hematological profile. How did the authors ensure that the changes in hematological profile between infected and uninfected participants can be attributed solely to the infection?

Line 112: Parasite load could affect whether or not the parasite can be seen in a given sample analyzed using microscopy. Was only a single fecal sample used for the study or multiple sampling per patient? Were the lab analyses done immediately after sample collection? Was there a need for storage? How was the sample stored? If the sample was also transported, the author should provide a brief description on how this was done.

Line 122: Was the slide examination performed by one microscopist? Was only a single slide prepared for each patient? Ideally there should be at least two. The authors should clarify if this was so and also indicate how inconsistent results were resolved.

Line 129: Authors should briefly describe which anthropometry was done. Dietary characteristics and nutritional status of pregnant women were also assessed. The authors should describe the tool used. If this was a questionnaire, the authors should indicate whether it was structured/non-structured or whether it was adapted from the WHO guideline. The authors should also be ready and willing to provide a copy of the questionnaire as per the journal’s requirements.

Results:

Table 2: It is possible that dietary characteristics of the women may change with time. The total indicates that the nutritional assessment reported is for first trimester. Did the authors assess the same for 2nd and 3rd trimesters?

Line 181-96 (Prevalence of intestinal parasite among pregnant women): Only the text has been provided. There is no table/figure to show any of the claims.

Line 197-202: The table referencing the effects of intestinal parasite on hematological profile in the text is table 3 (line 202) which rather provides information on MUAC feature of the pregnant women.

Table 4 and 5 should be merged.

Table 4: The authors appears to have combined 1st, 2nd and 3rd trimester data in this test. This will mean that each patient (minus 1) will have 3 inputs for a given data. The authors should revisit this analysis, separating 1st, 2nd and 3rd trimester data and possibly providing analysis for each. With this, it will be possible to uncover information about which stage of the pregnancy that parasitic infection most relate to anemia in the study region.

Line 206-210 appear to talk about the use of logistic regression to identify which parasite has higher odds ratio for anemia but the associated table seem to be a continuation of table 4 (A t test). The authors should check and rectify.

Line 212-226: There is no table/figure to support this text.

Discussion:

Line 233: The claim that there are few studies that examine the burden of intestinal parasite in pregnant women is debatable. There are in fact a lot of studies even in Ethiopia alone. The authors should rephrase it.

The authors make claims about possible reasons that underlie the differences in findings when compared with other studies (e.g., line 260-61: “The difference may be due to study area, population and life style that vary from state to state.) but fail to provide adequate evidence on what the underlying differences are. How is the study area and population different? What is the lifestyle of the study area that impact the finding but is absent in the study being compared to?

A major weakness of the discussion is the lack of synthesis on the public health implications (the “and so what” part of the findings). The discussion and conclusion need to be strengthened.

The author claim that the study was a prospective one. It will be of great importance if the author could also provide data on the possible impact of the parasitic infection and the anemia on the birth weight and other clinically relevant data from the babies.

The study also has numerous limitations which the authors should indicate. For example, the sample size, although calculated, remains small. Several underlying conditions that could affect the hematological profile were not tested. Although microscopy is useful for detecting parasitic infections, it still presents with substantially low sensitivity that will lead to missed cases especially in the presence of low intensity infections, etc.

Minor issues:

Although the study seems to be thorough, the language is unclear which makes it difficult to follow. The authors, with the help of the journal, could improve the language to make it easier for the readers.

The authors should also avoid repeating the results (numbers and percentages) in the discussion. Paraphrasing the main findings will be beneficial.

The authors may also want to also revisit the referencing of texts. Some statements are made (such as line 623) without a reference.

The authors may want to avoid starting sentences with numbers. There is also no need to be stating the total of each variable in table 1 if all sum up to the overall total. “Not read and write” can also be changed to “No formal education”.

Line 102-3 “Pregnant women who attend antenatal care (ANC) unit at Debre Markos referral hospital during the study period” appears incomplete.

Reviewer #6: Dear PloS One editor,

Here are the general comments for the author. The comments are briefly described in the main manuscript and high-lighted using yellow texts and accessible to the author in charge of the manuscript for reaction.

To highlight some of the comments,

Abstract

To some extent, it doesn’t reflect the main manuscript, especially of methods and result part. So, it needs re-construction.

Methods

o Study design: requires justification for study design to be cohort and prospective. The result gives some clues for it to be more likely cross-sectional type and retrospective despite of the long data collection period

o The sample size is not known and it lacks calculation for computing sample size

o The sampling technique is not stated

o The issue of ethical approval needs evidence like approval number should be stated or approval letter should be attached/submitted

Result

o In general, the organization is poor; statistical tools are not well described especially the write-up for example while writing OR with 95% C.I and P-value

o There is not table that describes multivariate analysis

o Table 2&3, their importance is under question in line with the study objective

o There are many grammatical and spelling errors in result description

o In table 5, it lacks descriptions concerning t-test result like for example the finding and its relevance

Discussion

o In summary, it lacks organized way of write-up and coherence. I kindly recommend improvement including grammatical and spelling errors corrections by using online grammar checker

Conclusion

o Some of it is not based on the result output. So, kindly correct it

References

o Citations are not according their order in the reference section

Reviewer #7: Although the authors addressed an interesting topic, the manuscript is very preliminary. In addition, there are problems on results presentation and and writing. Besides that the results and the conclusion cannot be supported by the results presented.

6. PLOS authors have the option to publish the peer review history of their article (what does this mean?). If published, this will include your full peer review and any attached files.

Reviewer #1: No

Reviewer #2: **Yes: **Tadesse Hailu

Reviewer #3: No

Reviewer #4: **Yes: **Ángela Fernanda Espinosa Aranzales

Reviewer #5: No

Reviewer #6: No

Reviewer #7: No

---

## [Author Response · Author response to Decision Letter 0]

25 Feb 2021

We had accepted and corrected all reviewers and editors comments.

---

## [Decision Letter · Decision Letter 1]

23 Mar 2021

PONE-D-20-36685R1

Effects of intestinal parasite infection on hematological profile of Pregnant Women Attending Antenatal care in Debre Markos Referral Hospital, North West Ethiopia. Institution based Prospective cohort study

PLOS ONE

Dear Dr. Demeke,

Thank you for submitting your manuscript to PLOS ONE. After careful consideration, we feel that it has merit but does not fully meet PLOS ONE’s publication criteria as it currently stands. Therefore, we invite you to submit a revised version of the manuscript that addresses the points raised during the review process.

We look forward to receiving your revised manuscript.

Kind regards,

Frank T. Spradley

Academic Editor

PLOS ONE

Journal Requirements:

Reviewers' comments:

Reviewer's Responses to Questions

**Comments to the Author**

1. If the authors have adequately addressed your comments raised in a previous round of review and you feel that this manuscript is now acceptable for publication, you may indicate that here to bypass the “Comments to the Author” section, enter your conflict of interest statement in the “Confidential to Editor” section, and submit your "Accept" recommendation.

Reviewer #1: All comments have been addressed

Reviewer #2: (No Response)

Reviewer #4: (No Response)

Reviewer #6: All comments have been addressed

2. Is the manuscript technically sound, and do the data support the conclusions?

Reviewer #1: Yes

Reviewer #2: (No Response)

Reviewer #4: Yes

Reviewer #6: Yes

3. Has the statistical analysis been performed appropriately and rigorously? 

Reviewer #1: Yes

Reviewer #2: (No Response)

Reviewer #4: Yes

Reviewer #6: Yes

4. Have the authors made all data underlying the findings in their manuscript fully available?

Reviewer #1: Yes

Reviewer #2: (No Response)

Reviewer #4: Yes

Reviewer #6: Yes

5. Is the manuscript presented in an intelligible fashion and written in standard English?

Reviewer #1: Yes

Reviewer #2: (No Response)

Reviewer #4: Yes

Reviewer #6: No

6. Review Comments to the Author

Reviewer #1: In this study the authors evaluated the effects of intestinal parasite infection on hematological profile of Pregnant Women.

It seems that the authors revised the manuscript according to the comments.

Reviewer #2: Authors have addressed many of my comments which was given previously but still the the grammar part is a critical problem throughout the documnet. So, much work should be done on the grammar issues.

Reviewer #4: Thank you for taking into account the most of my comments and for your hard work. However, It still is necessary to address some few comments. Please check in the attachment file.

Reviewer #6: Despite addressing majority of the comments, still a few comments to the author:

1. The tables in general are poorly organized. So. I kindly ask the author to organize the lay out of the tables so that it can fit to one page preferably. Minimize the space between the rows.

2. There noticed grammatical error for example in line 199. It says " didn't taken

7. PLOS authors have the option to publish the peer review history of their article (what does this mean?). If published, this will include your full peer review and any attached files.

Reviewer #1: No

Reviewer #2: **Yes: **Tadesse Hailu

Reviewer #4: **Yes: **Ángela Fernanda Espinosa Aranzales

Reviewer #6: No

---

## [Author Response · Author response to Decision Letter 1]

31 Mar 2021

Thank you for your comments.We have see the document detail and all comments are corrected and included in the manuscript and in the response letter.

---

## [Editor Report · Decision Letter 2]

6 Apr 2021

PONE-D-20-36685R2

Effects of intestinal parasite infection on hematological profile of Pregnant Women Attending Antenatal care in Debre Markos Referral Hospital, North West Ethiopia. Institution based Prospective cohort study

PLOS ONE

Dear Dr. Demeke,

Thank you for submitting your manuscript to PLOS ONE. After careful consideration, we feel that it has merit but does not fully meet PLOS ONE’s publication criteria as it currently stands. Therefore, we invite you to submit a revised version of the manuscript that addresses the points raised during the review process.

ACADEMIC EDITOR COMMENT: It seems as if the authors have responded to most of the reviewers' comments. However, an issue still stands with English grammar and syntax. For instance, there are words than need hyphenating, like "intestinal parasite-free pregnancies" and there are several incomplete sentences, among other issues. You must contact a copyeditor/colleague that has expertise in the area of English writing and composition before resubmitting this manuscript.

We look forward to receiving your revised manuscript.

Kind regards,

Frank T. Spradley

Academic Editor

PLOS ONE
---

## [Author Response · Author response to Decision Letter 2]

13 Apr 2021

Dear all; All comments and corrections were made.

---

## [Editor Report · Decision Letter 3]

19 Apr 2021

Effects of intestinal parasite infection on hematological profile of Pregnant Women Attending Antenatal care in Debre Markos Referral Hospital, North West Ethiopia. Institution based Prospective cohort study

PONE-D-20-36685R3

Dear Dr. Demeke,

We’re pleased to inform you that your manuscript has been judged scientifically suitable for publication and will be formally accepted for publication once it meets all outstanding technical requirements.

Kind regards,

Frank T. Spradley

Academic Editor

PLOS ONE

---

## [Editor Report · Acceptance letter]

29 Apr 2021

PONE-D-20-36685R3 

Effects of intestinal parasite infection on hematological profiles of pregnant women attending antenatal care at Debre Markos Referral Hospital, Northwest Ethiopia: Institution based prospective cohort study 

Dear Dr. Demeke:

I'm pleased to inform you that your manuscript has been deemed suitable for publication in PLOS ONE. Congratulations! Your manuscript is now with our production department. 

Kind regards, 

on behalf of

Dr. Frank T. Spradley 

Academic Editor

PLOS ONE